# Mitochondrial Heterogeneity in Metabolic Diseases

**DOI:** 10.3390/biology10090927

**Published:** 2021-09-17

**Authors:** Jennifer Ngo, Corey Osto, Frankie Villalobos, Orian S. Shirihai

**Affiliations:** 1Department of Medicine, Division of Endocrinology, David Geffen School of Medicine at UCLA, 650 Charles E. Young Drive East, Los Angeles, CA 90095, USA; jngo@chem.ucla.edu (J.N.); costo@mednet.ucla.edu (C.O.); frankyv16@g.ucla.edu (F.V.); 2Department of Chemistry and Biochemistry, University of California, 607 Charles E. Young Drive East, Los Angeles, CA 90095, USA; 3Metabolism Theme, David Geffen School of Medicine, University of California, Los Angeles, CA 90095, USA; 4Department of Molecular and Medical Pharmacology, David Geffen School of Medicine at UCLA, 650 Charles E. Young Drive East, Los Angeles, CA 90095, USA; 5Department of Integrative Biology and Physiology, University of California, Los Angeles, CA 90095, USA; 6Molecular Biology Institute, University of California, Los Angeles, CA 90095, USA

**Keywords:** mitochondria, heterogeneity, subpopulations, lipotoxicity, type 2 diabetes, kidney diseases, morphology, membrane potential, calcium

## Abstract

**Simple Summary:**

Often times mitochondria within a single cell are depicted as homogenous entities both morphologically and functionally. In normal and diseased states, mitochondria are heterogeneous and display distinct functional properties. In both cases, mitochondria exhibit differences in morphology, membrane potential, and mitochondrial calcium levels. However, the degree of heterogeneity is different during disease; or rather, heterogeneity at the physiological state stems from physically distinct mitochondrial subpopulations. Overall, mitochondrial heterogeneity is both beneficial and detrimental to the cellular system; protective in enabling cellular adaptation to biological stress or detrimental in inhibiting protective mechanisms.

**Abstract:**

Mitochondria have distinct architectural features and biochemical functions consistent with cell-specific bioenergetic needs. However, as imaging and isolation techniques advance, heterogeneity amongst mitochondria has been observed to occur within the same cell. Moreover, mitochondrial heterogeneity is associated with functional differences in metabolic signaling, fuel utilization, and triglyceride synthesis. These phenotypic associations suggest that mitochondrial subpopulations and heterogeneity influence the risk of metabolic diseases. This review examines the current literature regarding mitochondrial heterogeneity in the pancreatic beta-cell and renal proximal tubules as they exist in the pathological and physiological states; specifically, pathological states of glucolipotoxicity, progression of type 2 diabetes, and kidney diseases. Emphasis will be placed on the benefits of balancing mitochondrial heterogeneity and how the disruption of balancing heterogeneity leads to impaired tissue function and disease onset.

## 1. Introduction

Cellular heterogeneity represents the observation that in an organ, neighboring cells of the same type may show structural and functional differences that evolve from their different stages of differentiation, maturation, or exposure to environmental challenges. Whichever is the trigger, the outcome is structural diversity that results in varying functionality. Similar to cellular heterogeneity, the pool of mitochondria within a singular cell may also represent mitochondria at different stages of their development as well as the consequences of their response to the cellular environment [1]. This review will focus on the subcellular heterogeneity of the mitochondria within the same cell, the mechanisms generating subcellular heterogeneity, and the functional consequences thereof [1,2,3,4].

Metabolic diseases can be inherited, known as inborn errors of metabolism, or they can be acquired. This review will focus on mitochondrial heterogeneity reported in metabolic diseases resulting from excess nutrient exposure, focusing on their effects in the pancreatic beta-cell and the renal proximal tubule cells. Specifically, an examination of the heterogeneity in mitochondrial membrane potential, morphology, and calcium reported in type 2 diabetes (T2D) and kidney diseases. Also included is a review of mitochondrial subpopulations that show the difference in their metabolism and function, but how these differences are generated remains unclear.

### 1.1. What Is Heterogeneity and Why Should We Care?

We define mitochondrial heterogeneity as the variation in a mitochondrial feature of a singular mitochondrion inside one cell. Amongst what has previously been considered as homogenous groups of mitochondria, higher-resolution technology continues to further the discovery of variation in ultrastructural, biophysical, and electrochemical features [2,5,6,7,8]. The observation that every biological structure will always show some level of diversity raises the question, when should we care? If mitochondrial membrane potential averages around −200 mV, should we care if some mitochondria are −199 mV and others are −201 mV? We suggest two considerations to determine if a variation in some parameter is of interest: either the heterogeneity yields functional consequences or the heterogeneity is an association with a phenomenon of interest. For example, mitochondria deficient in cristae heterogeneity carry the functional consequences of reduced respiratory capacity [7,9,10]. Mitochondrial fragmentation has been associated with stress, yet direct functional consequences of the heterogeneity introduced by fragmentation have not been fully determined. For this review, we will describe the observed variations and functional consequences in mitochondrial ultrastructure and dynamics associated with metabolic diseases, T2D and chronic kidney diseases. It is not to say heterogeneity only occurs under pathological states; rather, heterogeneity is heightened. Indeed, some level of heterogeneity is physiological. Mitochondrial variance at the physiological state provides functional fitness to specific groups of mitochondria within the cell; yet, upon chronic stress, physiological levels of heterogeneity begin to shift towards pathological levels [11]. For instance, a small percentage of mitochondria undergo depolarization as they prepare to be removed by mitophagy; however, under the pathological impairment of mitophagy, the percentage of depolarized mitochondria increases [7,12]. Impaired clearance and accumulation of depolarized and damaged mitochondria lead to the release of apoptogenic factors and cellular apoptosis. Whether changes to mitochondrial features are small or large, mitochondrial heterogeneity matters when changes yield functional consequences or are associated with a phenomenon of interest.

### 1.2. Nongenetic Contributors to Mitochondrial Homogeneity and Heterogeneity

Mitochondrial heterogeneity can be caused by genetic and non-genetic mechanisms. Variation in mitochondrial DNA (mtDNA) sequences provides genetic sources of intracellular heterogeneity, influencing factors such as reactive oxygen species (ROS) production, protein structure conformation, and electron transport chain capacity [13]. However, structural and functional variation is also controlled by post-translational mechanisms such as phosphorylation and ubiquitination of mitochondrial proteins. Further, mitochondrial heterogeneity can be established by the molecular machinery determining the mitochondrial life cycle: biogenesis, mitochondrial motility, fusion and fission, and clearance of damaged mitochondria. Sustained activity of the mitochondrial life cycle serves to decrease mitochondrial heterogeneity (Figure 1A). However, blockades at different points of the mitochondrial life cycle may induce heterogeneity by expanding a specific stage of the life cycle disproportionally (Figure 1B).

A higher number of fusion events promote homogeneity by increasing the rate by which different mitochondria within a cell mix and equilibrate mitochondrial content [12]. Impaired fusion reduces the rate of mitochondrial content mixing and increases heterogeneity. Depolarization of an individual mitochondrion induces the recruitment of fission mechanisms. The impaired portion of a mitochondrion can be removed through fission into two daughter mitochondria. One of the daughter mitochondria will come out of the fission event with reduced membrane potential. If membrane potential does not recover, the mitochondrion will sustain depolarization and remain solitary until it is removed by mitophagy. During this period, this mitochondrion remains depolarized and fusion-less, two characteristics of the mitochondrial pre-autophagic pool. Autophagic clearance of mitochondria reduces heterogeneity by removing depolarized mitochondria and promotes homogeneity by maintaining a pool of similarly polarized mitochondria. Thus, the progression at the fission and mitophagy stages influences mitochondrial heterogeneity (Figure 1). Mitochondrial heterogeneity in this sense may seem to be detrimental to cellular function; however, observations have shown mitochondrial heterogeneity to be beneficial and to play an important role in cellular fitness [7,14]. Additionally, the long-term consequences of declining heterogeneity lead to disease onset [15,16]. Through the use of MitoTimer, it was revealed that the age of mitochondrial protein is homogenous across the cell. Yet, inhibition of mitochondrial fusion resulted in the appearance of two mitochondrial populations, mitochondria primarily containing old protein and mitochondria containing primarily young protein. Such heterogeneity based on protein age demonstrated the role of mitochondrial fusion in homogenizing the distribution of newly synthesized proteins [15].

In this review, we focus on literature concerning mitochondrial heterogeneity in pancreatic beta-cells and proximal tubules. Specifically, how mitochondrial heterogeneity fluctuates in response to nutrient excess and affects cellular behavior at the pathological state, how physiological heterogeneity exists between mitochondrial subpopulations, and finally how physiological heterogeneity contributes to mitochondrial function.

## 2. Mitochondrial Heterogeneity Increases under Pathological States

Mitochondria are highly dynamic organelles that respond to changes in their immediate environment through diverse structural and functional adaptation, resulting in intracellular heterogeneity. When subcellular heterogeneity is induced by metabolic stress, we define this occurrence as the pathological state, under which mitochondria display an increase in functional diversity. In pancreatic beta-cells and renal proximal tubule cells, metabolic stress via high levels of glucose and free fatty acids increase heterogeneity by disrupting the mitochondrial fusion. Depending on the degree and duration of metabolic stress, increased heterogeneity affects cellular behavior and can contribute to the pathogenesis of T2D and chronic kidney disease [16,17,18].

### 2.1. Membrane Potential Heterogeneity Reflects Diverse Mitochondrial Response to Cellular Nutrient Load

Mitochondria within the same cell display a wide heterogeneity in mitochondrial membrane potential (MMP) (Figure 2A) [9,19,20]. The range of MMP variation is considerably larger when challenged by stressors such as oxidative stress, aging, and excess fuel, suggesting that pathology can be associated with higher heterogeneity in MMP. In hepatic, cardiac, and neuronal cell models, it has been noted that mitochondria depolarize or hyperpolarize in the face of stressors, and not all mitochondria respond similarly to stress [19,21,22,23]. As an example, in cardiomyocytes, subcellular domains with elevated ROS or changes in pH induce depolarization in neighboring mitochondrial subsets, by activating the permeability transition [24]. In hepatocytes isolated from old rats, Nicholls et al. showed mitochondria in the same cell did not all have the same MMP. Differences in voltage were observed in one mitochondrial population: 10% of the mitochondria had an MMP of 154 mV, 65% had 94 mV, and 25% had 74 mV [19]. The observed diversity in MMP is in part attributed to factors such as the subcellular localization of mitochondria within a hepatocyte, with these locations showing differences in fuel supply and ATP demand. As a result of differential substrate and ion exposure, shifts in ATP demand and fuel availability generate fluctuations in MMP. Glucose availability influences the metabolic state of the cell and contributes to rapid MMP fluctuations [20].

An excessive supply of glucose and free fatty acid (FFA) levels increased intracellular MMP heterogeneity in beta-cells, conditions mimicking T2D in vitro [5,21,24]. This effect of high nutrient availability in vitro supports that increased MMP heterogeneity within a single beta-cell is a characteristic of mitochondrial dysfunction in diabetic beta-cells. Single beta-cells treated with exogenous substrates saw a 37% increase in MMP heterogeneity compared to untreated cells—all the while maintaining morphology [5,24]. Hyperpolarization of beta-cell mitochondria has been well correlated with insulin secretion in response to glucose concentrations [20]. Upon assessing contributors to the mechanism of glucose-induced MMP heterogeneity, Wikstrom et al. identified BAD, a proapoptotic BCL-2 family member. This identification suggested that individual mitochondria may carry different amounts of BAD protein to generate MMP heterogeneity. Plasma membrane depolarization mediated by ATP closing K_ATP_ channels and an increase in cytosolic calcium are both shown to be triggering events in glucose-stimulated insulin secretion [25]. As such, glucose-induced mitochondrial hyperpolarization functionally affects the glucose concentration-dependent increase in insulin secretion [25,26].

Similar to beta-cells, kidney diseases also yield disruption of homogenous intracellular MMP. Increases in kidney MMP heterogeneity have been observed to either increase due to ROS, a decline in ATP production, or loss of cristae structure [27,28,29]. Specifically, changes to membrane potential have been observed in these two kidney disease models: (1) reduction in FAO and (2) loss of AMPK activity. In the first case, impaired FAO was modeled by the knockdown of the FAO enzyme CPT2. The CPT2 knock-down model reduced mitochondrial FAO, reduced lipid accumulation, increased renal fibrosis, and increased intracellular MMP heterogeneity. CPT2-deficient mitochondria measured by JC-1 revealed significant increases in membrane potential standard deviation, a measure of intracellular membrane potential heterogeneity [28]. In a separate study, Kodiha et al. developed a model for kidney disease by inhibiting AMPK activity in LLC-PK1 cells, cells derived from the renal proximal epithelium. Short-term inhibition of AMPK resulted in both mitochondrial fragmentation and significant increases (+32%) in MMP as measured by MitoTracker CMX ROS/TOM70 ratio. Furthermore, treatment with compounds aimed at AMPK modulation also resulted in increased MMP heterogeneity. Some compounds elevated intracellular MMP overall while maintaining differences in morphology; others reduced intracellular MMP heterogeneity across the mitochondrial population [29]. The variation in increasing or reducing membrane potential observed across treatments leads authors to postulate the variance, as a readout either as drug effectivity or as membrane potential may influence treatment effects. In both disease models, cellular metabolism was reflected in MMP, and a pathological decline in proximal tubular FAO and lipid accumulation was reflected in increased intracellular MMP heterogeneity.

### 2.2. Brief Overview of Mediators in Mitochondrial Fusion and Fission

Mitochondrial architecture is determined by motility, fusion, and fission events. Although some disorders of mitochondrial dynamics result from monogenic mutation, most reflect changes in the function of fission and fusion mediators due to changes in the cellular environment. Fission and fusion influence cellular processes such as ATP production, ROS generation, and calcium homeostasis and consequently impact bioenergetics and mitophagy [30].

Fusion in mammals is mediated by the outer mitochondrial membrane proteins Mitofusin 1 and 2 (Mfn1 and Mfn2) and the inner mitochondrial membrane protein, optic atrophy gene 1 (Opa1). Mitofusins are targeted to the mitochondria by sequences in their transmembrane and C-terminal domains. Through the C-terminal, Mfn1/2 initiate fusion by creating homodimeric or heterodimeric, antiparallel, coiled-coil linkages between mitochondria. Mfn2 is also located in the endoplasmic reticulum (ER) and promotes ER-mitochondria tethering. Calcium storage is one of the functions commonly attributed to the ER; hence, mitochondria–ER tethering enhances mitochondrial calcium uptake and alters morphology. In the case of OpaI, fusion activity is mediated by proteolytic processing. OpaI also controls cristae remodeling independent of fusion [30,31,32]. OPA1 has eight splice variants, each with differential fusion activity and mitochondrial protease susceptibility. The fusion mediators also regulate mitochondrial metabolism, and when they are downregulated or dysfunctional, there is generally a reduction in mitochondrial oxidative capacity [10].

On the other hand, mitochondrial fission is mediated by the outer mitochondrial membrane proteins, fission 1 protein (Fis1) and mitochondrial fission factor (Mff), and the cytosolic dynamin-related protein 1 (Drp1). GTP hydrolysis and recruitment of Drp1 to the outer mitochondrial membrane are required for Drp1-mediated fission [33]. Furthermore, Drp1 activity is regulated by the phosphorylation of serine 616 and serine 637. Phosphorylation of serine 616 increases DRP1 activity, whereas phosphorylation of serine 637 decreases it. Mff and Fis1 have been shown to mediate fission by recruiting Drp1 to the mitochondria [10,34,35].

### 2.3. Architectural Variance under Pathological States Introduces Metabolic Defects

Mitochondrial function and morphology are closely linked; the mitochondria life cycle involves changes to morphology to preserve mtDNA integrity and mitochondrial bioenergetic efficiency. Increases in intracellular mtDNA heterogeneity (i.e., accumulation of damaged mtDNA) and declines in fusion/fission regulation result in tissue dysfunction [36].

Increased intracellular heterogeneity in morphology has been observed under various metabolic states, such as glucotoxicity, oxidative stress, and starvation (Figure 2B). Under glucotoxicity-induced ROS, enhances in mtDNA oxidation and mitochondrial fragmentation result in an increased pool of fragmented mitochondria. Skeletal muscles under oxidative stress yield increased architectural heterogeneity due to a 30% reduction in mitochondrial velocity. Additionally, reductions in fission and fusion rates led to a 41% increase in fragmentation [37]. During starvation, mitochondria unable to elongate are latently dysfunctional and consume cytosolic ATP to sustain their membrane potential. The survival of non-elongated mitochondria suggests a favoring for elongation; however, survival of fragmented mitochondria increases intracellular mitochondrial architectural heterogeneity [38]. These three cases demonstrate increased architectural heterogeneity under nutrient stress. Apart from the mitochondrial genome, metabolic stress or pathology increase morphological heterogeneity at the level of fusion and fission rates.

In the pancreatic beta-cell, mitochondria go through frequent fusion and fission events that are changed by altered nutrient exposure [39]. Thus, fission and fusion are strong modifiers of mitochondrial heterogeneity. Under pathological conditions such as T2D, the balance of fusion and fission leans in favor of fission and reduction in fusion [24,40]. Beta-cells are susceptible to chronic exposure to high glucose levels, a condition termed glucotoxicity. Glucotoxicity has been shown to inhibit mitochondrial fusion and induce mitochondrial fragmentation. Further increasing the duration of exposure to high glucose concentrations exacerbates the degree of architectural heterogeneity so much that the degree of changes becomes pathological fragmentation. As glucose concentration increases, there is a reduction in the mitochondrial fission protein, Fis1. In this pathological state, morphology is altered and fragmentation occurs whereby Fis1 impairs ATP production and insulin secretion [12,26,41]. Under glucolipotoxic conditions, high glucose and high fatty acid levels, beta-cell mitochondria become more fragmented, resulting in reduced glucose-stimulated insulin secretion [24]. Although there was an overall average reduction in the fusion rate of treated cells, fusion rates varied both intercellularly and intracellularly. Fusion rate was measured by the diffusion rate of GFP in mitochondria expressing PAGFPmt. Focusing specifically on intracellular measurements, mitochondria with more branched networks had greater fusion rates than less branched mitochondria. At the time, mitochondria reported to perform fusion at higher rates were termed super fusers [24]. More recent literature suggests super fusers and mitochondria of reduced fusion may be two separate populations of mitochondria within the same cell (further discussed in Section 3). Another established feature of T2D is reduced circulating insulin levels due to decreased beta-cell mass caused by beta-cell apoptosis [42,43]. In a cultured beta-cell model, high glucose concentrations induce the functional outturn of apoptosis; however, inhibition of mitochondrial fission prevents beta-cell apoptosis [44]. Reiterating fission and fusion are strong modifiers of mitochondrial heterogeneity and the consequences thereof.

Mitochondrial fragmentation is not only limited to the pancreatic beta-cell; increased fission has also been observed in kidney biopsies of diabetic human patients [18]. Recent work modeling renal ischemia-reperfusion, nephrotoxicity, and hyperglycemia-induced kidney injury have all demonstrated increased mitochondrial fragmentation [18,45,46]. The above studies suggest increased morphological heterogeneity as an associated phenomenon. Specifically, intracellular mitochondrial fragmentation has been observed to be associated with renal pathology. Different pharmacological approaches have been used to overcome the loss of mitochondrial performance in kidney diseases; of these, a predominant target is the Ser/Thr protein kinase 5′AMP-activated kinase (AMPK). Activation of AMPK downregulates ATP-consuming pathways and shifts the metabolic state towards catabolism, and long-term activation yields mitochondrial biogenesis [29]. In the clinical setting, AMPK activity is enhanced by the antidiabetic drug metformin. To study the pathway more in the context of kidney disease, Kodiha et al. induced pathology through AMPK inhibition. The pathological model elicited fragmentation; however, intracellular morphological changes were not homogenous—rather, the degree of fragmentation differed by subcellular region. For the most part, the degree of architectural change was compound-specific and observed to correlate with mitochondrial compartmentalization [29].

Increased intracellular fragmentation was associated with the pathological model of AMPK inhibition. If inhibiting intracellular fragmentation is no longer feasible, is it still feasible to ameliorate pathology by reducing intercellular fragmentation? In both 2D and 3D EM micrographs of control tubular cells, a large intercellular population of long and filamentous mitochondria was observed. In sharp contrast, mitochondria of tubular cells injured by ischemia/reperfusion were completely fragmented. Looking broadly at the intercellular population, ischemia/reperfusion fragmented 42% of the cellular population [17]. Upon pharmacological blockade of mitochondrial fission in ischemically injured cells, attenuation of tubular cell apoptosis, tissue damage, and renal injury was observed [17]. Perhaps intracellular heterogeneity occurs in the early stages of pathology, and once a threshold has been surpassed intercellular heterogeneity is reached. Mitochondrial fragmentation has been observed to occur early in pathology and contributes to the subsequent development of mitochondrial membrane permeabilization, release of apoptogenic factors, and cellular apoptosis [47]. Inhibition of intercellular mitochondrial fragmentation has been observed to protect against tubular cell apoptosis and renal injury. Benefits in reversing morphological changes suggest the pathological shortcomings of increasing mitochondrial heterogeneity both intracellularly and intercellularly. Thus, mediators of fusion and fission would identify possible targets for preventing and treating kidney diseases, such as acute renal failure [17].

Kidney function depends on oxidative metabolism to support its high energy demand to maintain electrolyte homeostasis, acid-base homeostasis, and reabsorption of nutrients. Energy depletion is a key contributor to the development and progression of kidney diseases including acute kidney injury (AKI) [45], chronic kidney disease (CKD) [46], and diabetic and glomerular nephropathy. Mitochondrial fatty acid oxidation (FAO) serves as the preferred fuel source supporting ATP production in the kidney, and its dysfunction results in ATP depletion and lipotoxicity [45,46,47]. FAO dysfunction elicits tubular injury, inflammation, and subsequent progression of fibrosis [46,48,49]. Miguel et al. assessed intracellular morphological alterations in cortical proximal tubule cells by transmission electron microscopy. At basal conditions, most mitochondria were elongated and localized to the basolateral part of the kidney cells. In contrast, in the 100 µM palmitate-treated primary mouse kidney epithelial cells, fewer mitochondria were localized to the basolateral region and exhibited a fragmented, small, and round appearance. Upon overexpression of the fatty acid shuttling enzyme, CPT1A, morphological alterations induced by fatty acids were abrogated in renal epithelial cells. The gain of function in FAO by CPT1A overexpression protected proximal tubules from fibrosis. In Miguel et al.’s study, defective FAO was observed in fibrotic kidneys from control mice of various fibrosis models, including the unilateral ureteral obstruction (UUO), folic acid nephropathy injury (FAN) model, and adenine-induced renal failure (AND). Overexpression of CPT1A in the mentioned fibrotic models counteracted impaired FAO and maintained comparable rates of FAO to that of healthy kidneys. The UUO, FAN, and AND models were all utilized to assess the benefits of recovering FAO at different stages whereby fibrosis occurred. The UUO models inflammation and fibrosis evident at 7 days due to obstruction to renal blood flow and glomerular filtration rate. The FAN models interstitial fibrosis in the chronic phase induced by high folic acid dosage. The AND model further complements renal failure models as a reflection of tubulointerstitial disease stemming from tubular toxicity of adenine metabolites. Miguel et al. postulate to preserve mitochondrial morphology, appropriate levels of CPT1A and metabolic function are necessary to maintain ATP demand [50]. Heterogeneity in this example is intriguing as the phenomenon of fragmentation is associated with the pathological impairment of reduced FAO through unknown mechanisms.

Under pathology, the balance between mitochondrial fusion and fission is disrupted, whereby fusion is reduced and the rate of fission is increased. The observed increases in intracellular fission occur to varying degrees, yet the outcome remains the same: altered mitochondrial function. Excess nutrient-induced mitochondrial fragmentation in islets significantly reduced insulin secretion. Additionally, an increase in the population of fragmented proximal tubule mitochondria consistently yielded reduced FAO. Overall, increased morphological heterogeneity is associated with metabolic defects seen in diabetic islets and kidney diseases (Figure 2).

### 2.4. Impaired Mitochondrial Calcium Buffering Drives Pathogenesis in Mitochondrial Signaling

Cytoplasmic calcium regulates several cellular functions including metabolism, transcription, and exocytosis [51]. Changes to cytosolic-free calcium directly affect intracellular calcium in organelles including the ER, mitochondria, secretory granules, and lysosomes [52]. At resting states, mitochondrial calcium yields similar concentrations to cytosolic calcium concentration. In this section, we will highlight functional outcomes observed in conjunction with heterogeneity in intracellular mitochondrial calcium levels under pathological states. As a key player in regulating the activity of several enzymes in the Krebs cycle, low mitochondrial calcium levels are reflected in low ATP production [53], whereas mitochondrial calcium overload invites apoptosis through the opening of the mitochondrial permeability transition pore (mPTP) [54,55]. The different pathologies arising from too high or too low calcium concentrations can occur in the same cell. Similarly, different impairments may occur in some mitochondria and not in others. This phenomenon can be explained by the concept that functional heterogeneity is induced by heterogeneous intracellular calcium levels. Changes to mitochondrial calcium levels in one subset of mitochondria and not another can be seen in diseases whereby not all mPTP are opened [56].

Indeed, disruption of cytoplasmic calcium microdomains has been observed under pathological states of skeletal, cardiac, beta-cells, and kidney cells. Calcium microdomains result in different calcium levels in the mitochondria. Under the pathological state, mitochondrial calcium levels are further exacerbated by the disruption of morphology and loss of mitochondrial–ER interactions. Impaired calcium signaling then manifests as increased resting calcium levels mediated by decreased activity of calcium transporters. In diabetic neurons, resting cytosolic calcium concentrations rise to 200 nM and trigger a rise in mitochondrial calcium buffering through mitochondrial calcium uniporter (MCU)-mediated uptake, resulting in partial mitochondrial depolarization. Diabetic neurons indirectly imply an increase in mitochondrial calcium heterogeneity in the form of prolonged calcium release.

In individual beta-cells, the kinetics of insulin secretion is determined by changes in cytoplasmic calcium [57]. Calcium is a regulator of insulin secretion and different mitochondria of various calcium levels may contribute unequally to secretion. Alteration of cytoplasmic calcium oscillations in beta-cells has been reported to be an early marker of islet dysfunction [58]. Animal models of obesity and T2D to some level also display disruption of cellular calcium influx [55,59,60,61,62,63], suggesting an association between increased intracellular calcium heterogeneity and metabolic dysfunction. Under pathological states of acute FFA versus glucolipotoxicity, calcium entry rates are divergent. Under acute FFA treatments, calcium influx is rapid, whereas under high glucose and palmitate levels calcium transport is delayed [64,65,66,67]. Heterogeneity in calcium influx rates may be explained by two possibilities: (1) changes to calcium transporter activity or (2) mitochondrial exposure to calcium microdomains. To address the first possibility, no changes were observed in the expression of the mitochondrial calcium transporters, MCU and NCLX [68]. As previously discussed in Section 2.3, the degree of intracellular heterogeneity in mitochondrial architecture varies between acute FFA treatment and GLT, suggesting mitochondrial calcium heterogeneity is not always influenced by transporter activity but may be influenced by mitochondrial morphology. The MCU’s low affinity to calcium promotes the formation of concentrated calcium microdomains on the surface of the mitochondria, which facilitates mitochondrial calcium uptake. Disruption of mitochondrial morphology and compartmentalization would disperse local calcium microdomains and ER–mitochondrial interactions. In the cases of acute FFA treatment and glucolipotoxicity, reductions in mitochondrial fusion and subcellular interactions via Mfn2 tethering to the ER have been postulated to contribute to functionally impaired mitochondrial calcium uptake (as discussed briefly in Section 2.2) [69,70]. Close associations, called mitochondria-associated membranes (MAMs), enable mitochondria to be in contact with high calcium microdomains [71,72]. Under chronic palmitate exposure, loss of local calcium microdomains and ER–mitochondrial interactions have been postulated to be the culprit of decreased insulin secretion [73,74].

The rise in intracellular calcium heterogeneity has been observed to influence mitochondrial morphology. In the Drp1 knockout beta-cell model, increased cytosolic calcium oscillations led to the aggregation of mitochondria to the nucleus and increased glucose-induced NAD(P)H across the cell [75], whereas Mfn1/2KO models prompted sharp reductions in glucose-induced cytosolic and mitochondrial calcium levels. Reductions in calcium levels in both compartments resulted in the functional consequence of impaired insulin secretion both in vivo and in vitro [76]. Impaired mitochondrial fission via Drp1KO and impaired fusion via Mfn1/2 KO in beta-cells demonstrate a link between calcium levels, mitochondrial localization and morphology, and beta-cell insulin secretion.

These findings then bring up the question, what promotes mitochondrial heterogeneity during the pathological state? From the reviewed literature, the pathological state is enabled by pre-existing levels of mitochondrial heterogeneity, the physiological state. We propose the heterogeneity observed in the pathological state is amplified by basal intracellular heterogeneity to induce functional compensation to cellular stress. The inability of mitochondria to adapt and recover or chronic exposure to cellular stress lead to decompensation and further development of disease.

## 3. Mitochondrial Heterogeneity as It Exists under Physiological States

Intracellular mitochondrial heterogeneity has been observed in physiological as well as pathological states. Mitochondria display heterogeneity in their morphology, membrane potential, and calcium levels; imaging at subcellular resolution correlated the diversity in the above parameters with specific subcellular localizations, leading to the hypothesis that intracellular mitochondrial heterogeneity is generated by mitochondrial subpopulations, as defined by their localization in the cell or by their proximity to other organelles [77,78,79]. Cardiac intermyofibrillar and subsarcolemmal mitochondria are some of the earliest examples of mitochondrial heterogeneity determined by their subcellular locations. The development of less crude mitochondrial isolation techniques enables the better study of not only their unique subcellular localization but also characterizing each subpopulation’s unique bioenergetics, proteome, lipidome, and molecular structure [79,80,81]. As of recently, the presence of mitochondrial subpopulations is no longer limited to cardiac and neuronal tissues but has also been observed in brown adipose tissues, proximal tubules, and pancreatic beta-cells [29,80,82,83]. Specifically, identified mitochondrial subpopulations have been termed perinuclear, peripheral, ER-bound, and peridroplet mitochondria (Figure 3). Despite observations of subpopulation-specific functions, it remains unclear if these unique functions stem from location-specific requirements or intrinsic factors. For example, cytoplasmic mitochondria have been identified to have their unique functional properties in the brown adipocyte; however, no consistent phenotype has been observed across “cytoplasmic mitochondria” of other tissue. For that matter, cytoplasmic mitochondria have been omitted from Figure 3.

How is it that intracellular mitochondrial heterogeneity can yield functional heterogeneity but not induce pathology? The lack of pathology may be explained by the hypothesis that mitochondria can form specific subpopulations that have complementary functions that serve their specific compartment. For example, mitochondria located in the vicinity of the nucleus promote communication between both the mitochondria and the nucleus, as well as maintaining higher levels of ATP/ADP and membrane potential [82]. At the cell periphery, mitochondria may support the function of ion pumps and protect against environmental insults through protective ion channels [1,88,89]. In this section, we will further describe mitochondrial heterogeneity observed under varying physiological states and address how mitochondrial subpopulations contribute to heterogeneity without contributing to the development of pathology in the context of the pancreatic beta-cell and renal proximal tubule.

### 3.1. Membrane Potential Heterogeneity Reveals Mitochondrial Subpopulations

At physiological states, membrane potential heterogeneity has been observed in the same mitochondrion. One end of the mitochondrion differs from either the midsection or the opposite end of the mitochondrion, a phenomenon termed mitochondrial discontinuity [9]. Mitochondrial discontinuity reveals the fission of a singular mitochondrion into two daughter mitochondria, both displaying different rates of depolarization. Groups of mitochondria displaying different depolarization rates promote physiological heterogeneity while supporting cellular function. When one group of mitochondria proceeds towards mitophagy, the other group maintains MMP and continues to support cellular function through ATP production. Mitochondrial subpopulations arise when mitochondrial discontinuity becomes compartmentally separated in the cell [4].

In non-diseased states, Aryaman et al. utilized JC-1 and observed intracellular MMP homogeneity in immortal cell lines from humans (Jurkat), fly (Kc167), and chickens (DT40) as well as primary rat hepatocytes and HUVECs [1]. On the other hand, Wikstrom et al. utilized TMRE and MitoTracker Green at non-pathological glucose levels (3 mmol/L glucose) and observed MMP heterogeneity. Beta-cells presented a standard deviation of 10.8 mV between mitochondrion in a single cell and 36% of the mitochondria in a cell were depolarized [5]. At 3 mmol/L glucose, beta-cells do not present secretion defects; however, the intracellular differences observed in mitochondrial membrane potential are enough to yield functional differences in ATP production. A difference of 7.1 mV in membrane potential translates to a five-fold difference in ATP synthesis capacity [19]. Heterogeneity in membrane potential and differences in ATP production may be explained by the existence of mitochondrial subpopulations in the beta-cell. Specifically, mitochondrial clusters surrounding the nuclei serve to drive ATP generation close to the nucleus to support energetic channeling and nuclear import [88,90]. Indeed, beta-cells have been observed to contain mitochondrial clusters surrounding the nuclei (Figure 3A). The existence of these perinuclear mitochondria in the beta-cell is suggested by the observation that perinuclear mitochondria display a stable membrane potential dissimilar to mitochondria not surrounding the nuclei [24]. The phenomenon of separate membrane potentials supported by subcellular compartmentalization suggests heterogeneity at the physiological state may be supported by the existence of mitochondrial subpopulations [8,26,91].

Similar to beta-cells, proximal tubule mitochondria also yield unique phenotypes linked to their distance from the nucleus. In LLC-PK1, cells derived from renal proximal epithelium, intracellular heterogeneity in MMP was observed at basal states when cells were divided into four regions radiating from the nucleus. Perinuclear mitochondria were defined as mitochondria within 1 µm of the nucleus and peripheral mitochondria were, at the outer most region, mitochondria localized to/near the cellular membrane (Figure 3B). MitoTracker CMX ROS/TOM70 values were higher in peripheral mitochondria compared to perinuclear mitochondria, revealing membrane potential heterogeneity associated with subcellular localization. Functionally, this would support the concept that those at the periphery are highly energized and potentially aid in sequestering more calcium ions [29,85]. At the periphery of proximal tubule cells, mitochondria with higher membrane potential would support transporters in the plasma membrane, as exemplified by Na+/K+- ATPase [92]. The localization-specific function would also support the hypothesis that mitochondrial subpopulations play a role in disease. Upon AMPK inhibition, LLC-PK1 perinuclear mitochondria increased in membrane potential and mitochondrial content, whereas the reverse occurred for peripheral mitochondria [29]. In an attempt to complement cellular needs, perinuclear mitochondria would reveal increased membrane potential as a reflection of altered functionality. As proposed for other cell types [78], ATP production in the vicinity of the nucleus could support energy-demanding processes, such as macromolecular trafficking through the nuclear pore [78] and the control of gene expression [82].

Overall, it has been observed in several studies that localization and organellular specific tethering by the mitochondria affect membrane potential and morphology, thereby influencing function. From this, we propose mitochondrial heterogeneity preexists due to the presence of subpopulations. Further, the observed heterogeneity induced by pathology is amplified intracellular heterogeneity. We further propose that the amplified intracellular heterogeneity favors the unique functions of one mitochondrial subpopulation over another mitochondrial subpopulation. The greater presence of one population over another is the cellular compensatory mechanism and improper balance of mitochondrial subpopulations leads to decompensation. Hence, there are pathological consequences of mitochondrial subpopulations and intracellular mitochondrial heterogeneity.

### 3.2. Mitochondrial Morphology Influences Metabolic Signaling and Nutrient Sensitivity

Mitochondrial morphology plays a key role in quality control both in the physiological and pathological states. In pathology, mitochondrial heterogeneity is largely observed in the context of reduced fusion and increased rates of fission. Under physiological states, intracellular architectural heterogeneity is observed between mitochondrial subpopulations. Subpopulations display different mitochondrial motility rates and hence the observation of differential fusion rates [80,93,94]. For example, using four-dimensional intravital microscopy, Porat-Shliom et al. observed mitochondrial motility differences in salivary gland cells based on subcellular localization. Salivary glands display mitochondria either compartmentalized to the basolateral plasma membrane or dispersed in the cytosol. At physiological states, cytosolic mitochondria exhibit microtubule-dependent motility and low fusion rates, whereas basolateral mitochondria are static and display high fusion rates. Upon β-adrenergic stimulation and exocytosis, cytoplasmic mitochondria increased motility and fusion events [75]. To verify if fusion rates are indeed affected by motility in cytoplasmic mitochondria, microtubule polymerization was inhibited, and motility and fusion specific to cytosolic mitochondria all saw reductions. By inhibiting cytoplasmic mitochondrial fusion reductions in the salivary gland, exocytosis was observed, suggesting mitochondrial subpopulation function plays a role in tissue functionality [95].

In a tomography study performed by Noske et al., intracellular and intercellular morphological heterogeneity at physiological states was observed in pancreatic beta-cells. To avoid extrinsic factors known to influence insulin secretion, the group assessed structural differences from two comparable beta-cells [76,96]. Beta-cells chosen were similar in both size and localization to the periphery of the pancreatic islet. Mitochondrial length differed between the two “equivalent” beta-cells as well as within the same cell. One beta-cell averaged 1770 nm with a 6% intracellular branching network and the other cell averaged 2300 nm in length with 10% mitochondrial populational branching. Heterogeneity can be observed intracellularly, as the tomography revealed mitochondria as long as 10,230 nm in a populational average of 1770 nm. Functionally, the group observed morphological differences that influence insulin secretion. Beta-cells with longer mitochondria revealed insulin granules to have been greatly discharged, suggesting morphological heterogeneity plays a role in glucose sensitivity even at physiological states [96].

In addition to revealing morphological heterogeneity at physiological states, the same study identified that a greater percentage of elongated mitochondria were associated with the Golgi ribbon [96]. Association with the Golgi ribbon was defined as mitochondrial clustering within 400 nm of the Golgi ribbon. Comparison of the two cells revealed greater Golgi ribbon–mitochondrial associations yielded greater glucose sensitivity [63,97,98]. Taken together, the study suggests greater glucose sensitivity and exocytosis are associated with mitochondrial branching and mitochondria greatly associated with or localized to the Golgi ribbon.

Tubular cells also demonstrate morphological heterogeneity at physiological levels [45,46,47,48,99]. In 2D EM images, tubular cells were composed of filamentous basolateral mitochondria and fragmented perinuclear mitochondria [17]. In mitochondria of ischemic tubular cells, all mitochondria were observed to be fragmented and no differences were observed between basolateral or perinuclear mitochondrial morphology. Inhibition of mitochondrial fragmentation has been observed to protect tubular cells from apoptosis and prevent renal injury. Suggesting a degree of fragmentation is adaptational, but fragmentation by all mitochondrial subpopulations is detrimental. Loss of a mitochondrial feature unique to a particular subpopulation, such as morphology, will either lead to failure to mediate pathological onset or sensitize the cell to environmental insults [17].

### 3.3. Mitochondrial Calcium as an Intracellular Heterogeneity Amplifier in Morphology and Motility

Subcellular calcium levels promote mitochondrial calcium heterogeneity at the physiological state in two manners, through dense calcium microdomains and calcium tunneling [100]. Subcellular calcium microdomains are found near calcium-rich storage centers such as the ER. Mitochondria located close to the ER or that are tethered to the ER typically yield greater levels of matrix calcium [94]. In the case of calcium tunneling, the mobilization of intracellular calcium through intracellular stores promotes calcium microdomains. Calcium tunneling can disrupt homogenous calcium concentrations in favor of dispersing calcium to other areas of the cell. Cells such as HeLa, PAEC, COS-7, HUVEC, hepatocytes, astrocytes, and neuronal cells have shown intracellular mitochondrial calcium heterogeneity to influence mitochondrial morphology, motility, and mitophagy [4,101,102,103,104,105,106]. For example, increased cytosolic calcium in cardiomyocytes and hepatocytes promotes mitochondrial fission [102,103]. In terms of motility, elevated calcium levels immobilize astrocytic mitochondria and low calcium levels induce mitochondrial mobility [104]. Similar motility behaviors to calcium have been observed in neurons [105,107]. Lastly, calcium sequestration and mitochondrial calcium overload promote mPTP activation, mitochondrial swelling, and mitophagy [54,93,101,108,109].

The ER is both the main calcium storage organelle and calcium supplier to the mitochondria [83,110,111]. Kyu Park et al. observed mitochondria capable of releasing calcium and “refilling” the ER in the absence of calcium-dependent currents. The calcium recycling phenomenon observed between mitochondria and the ER was only seen in mitochondria bound to the ER. Calcium recycling events were absent in mitochondria not associated with the ER [77], suggesting a specialized subpopulation of mitochondria bound to the ER (Figure 3C) [4].

It was revealed by Cardenas et al. that calcium transfer from the ER to the mitochondria is essential to maintain mitochondrial energetics [111]. Disruption of mitochondrial tethering to the ER via loss of Mfn2 yielded bioenergetic defects and decreased mitochondrial calcium levels [112]. Loss of Mfn2 has been attributed to loss of membrane potential and fragmentation in morphology, features of which can affect calcium retention and tethering. Subsequently, recovery of these characteristics did not recover calcium levels nor did they recover mitochondrial bioenergetics. However, recovery of the interaction between the two organelles revealed recovery of mitochondrial calcium levels as well as bioenergetic readouts. Heterogeneity in intracellular calcium metabolism further supports the existence of a specific subset of mitochondria bound to the ER [84,104,105,106,111,112].

### 3.4. Mitochondrial Subpopulations Influence Cellular Function

Mitochondrial heterogeneity in both pathological and physiological states has been observed to yield functional heterogeneity, if not an association with a phenomenon of interest. Intracellular mitochondrial heterogeneity in the physiological state has largely been attributed to the presence of various subpopulations (Figure 3). When mitochondrial subpopulations begin to lose their unique traits, cellular function is altered, and early stages of pathology are observed (as discussed in Section 3.1 and Section 3.2). The detrimental decline in mitochondrial subpopulation heterogeneity can explain the loss of compensatory mechanisms (as reviewed by Shum et al. [79]). Observation of disease onset due to loss of organellular heterogeneity has also been observed in lipid droplets [14]. This section will highlight known mitochondrial subpopulations and their influence on cellular function.

Among the more characterized mitochondrial subtypes are the cardiac intermyofibrillar and subsarcolemmal mitochondria. Initially characterized by ultrastructural differences, the two are physically separated, each revealing their biochemical functions. In the intermyofibrillar mitochondria, succinate dehydrogenase and citrate synthase activities were elevated compared to subsarcolemmal mitochondria. Additionally, calcium uptake capability differed and oxidation of substrates was 1.5× faster in intermyofibrillar isolates [2]. Given the complexity and diversity of the mitochondrial proteomic profiles, it is not surprising that mitochondria perform a diverse spectrum of functions to adapt to cellular needs [77,78,79,84,113].

In the case of brown adipose tissue, Benador et al. observed diverging roles between peridroplet mitochondria (Figure 3D), those attached to the lipid droplet, and cytoplasmic mitochondria. Specifically, peridroplet mitochondria (PDM) isolates have increased pyruvate oxidation, increased ATP synthesis capacity, and the function to support lipid droplet expansion via triacylglyceride synthesis, whereas cytoplasmic mitochondria undergo fatty acid oxidation. Interestingly, these two populations remain segregated, as PDM have their unique fusion-fission dynamics, preventing PDM from fusing with cytoplasmic mitochondria and homogenizing the mitochondrial population. The lack of fusion between PDM and cytoplasmic mitochondria thereby sustains two distinct mitochondrial proteomes in the same cell [80]. However, when brown adipocytes are required to switch fuel utilization to solely FAO, peridroplet mitochondria are observed to depart from the lipid droplet [114]. The transition of one mitochondrial subpopulation to another as a cellular compensatory mechanism supports the hypothesis that mitochondrial subpopulations influence cellular function.

### 3.5. Mitochondrial Heteroplasmy Potentially Promote Formation of Mitochondrial Subpopulations

Intracellular mitochondrial heterogeneity at physiological states has been observed to stem from global architectural changes and mitochondrial subpopulations, but ultimately how are mitochondrial subpopulations generated? What are some contributing factors to inducing mitochondrial heterogeneity? Here, we discuss three potential players: biogenesis, architectural adaptations, and subcellular localization (Figure 4). Biogenesis and subpopulation shifts can be explained by the concept of mtDNA expressing the “fittest” mtDNA to complement the cellular environment. Heterogeneity of mtDNA, known as heteroplasmy, has been observed to play a role in the biogenesis of mitochondria complementing cellular systems through bidirectional communication between the mitochondria and the nucleus (Figure 4A) [107]. In humans, mtDNA carries 37 genes, encoding 22 tRNAs, 2 rRNAs, and polypeptides of the electron transport chain complexes all within 16,569 bp. However, mitochondrial protein composition is estimated to approach 1500 proteins encoded by the nuclear genome [115]. With the vast composition, only 57% of mitochondrial proteins have been identified to be consistent between brain, kidney, liver, and heart mitochondria—the remaining 43% give rise to cell-type-specific differences in mitochondrial function [113].

Additionally, due to the lack of protective histones, mtDNA is more vulnerable to oxidative stress than nuclear DNA [116,117,118]. However, mitochondrial elongation has been observed to enhance resistance to oxidative stress [119], placing elongated mitochondria at the favored position of maintaining their pool of mtDNA for further biogenesis. Mitochondria absent of ultrastructural adaptations thereby lose their pool of mtDNA (Figure 4B). These morphological adaptations, if chronically maintained, may lead to changes in intracellular mitochondrial heteroplasmy [18]. Structural, functional, and behavioral heterogeneity then explain how multiple subpopulations have been identified in the mitochondria (Figure 4C). Hence, mitochondrial biogenesis from various mtDNA can lead to the formation of mitochondrial subpopulations.

As discussed above, mitochondrial biogenesis, architectural adaptations, and subcellular localization are some key players in inducing mitochondrial heterogeneity and generating unique subpopulations. Upon changes to the cellular milieu, specific mitochondrial subpopulations are favored to meet energetic demand. For example, increased demand for FAO in the brown adipocyte leads to the detachment of PDM and a general increase in the cytosolic mitochondria population [75]. The shifting of PDM to cytoplasmic mitochondria can be seen as a reduction in intracellular heterogeneity. Through what processes would intracellular heterogeneity be reduced in the presence of mitochondrial heteroplasmy? Some potential players in reducing mitochondrial heterogeneity are bottleneck heteroplasmy, mitochondrial fusion, and fission and mitophagy (Figure 5). Genetic bottlenecks lead to the rapid segregation of variants and if the variant takes over the population of mtDNA, homoplasmy results (Figure 5A) [120]. Contributors to bottleneck events may include mitochondrial fission or mitophagy (Figure 5B,C). While bottlenecking is not associated with a disease state at all times, to a certain degree, loss of heteroplasmy can limit metabolic flexibility. In the case of mitochondrial fusion, heterogeneity is diminished by the mixing and equilibrating of mitochondrial contents from different mitochondria (Figure 5B). As heteroplasmy increases, so too does the degree of mtDNA mutations [120]. Within the pool of mitochondria selected to be removed by mitophagy are damaged mitochondria and mitochondria carrying deleterious mutations in mtDNA (Figure 5C) [120,121,122]. Overall, genetic bottlenecks, fusion and fission, and mitophagy may reduce intracellular heterogeneity by removing variants encoding various mitochondrial subpopulations.

Though mitophagy may reduce metabolic flexibility by eliminating a variant encoding a mitochondrial subpopulation, mitophagy as a reducer of mitochondrial heterogeneity (Figure 5C) has largely been associated with better disease alleviation [123,124,125,126]. Reduced de novo mitochondrial production, accumulation of damaged mitochondria, and reduced expression of mitophagy genes have all been contributors to mitochondrial dysfunction [123]. The use of compounds such as urolithin A, AMPK activators, NAM, and actinonin, all of which reactivate mitophagy, has been shown to replenish cells with mitochondria of higher quality and alleviate disease symptoms [123,124,125,126]. In patients of Duchenne muscular dystrophy (DMD), the administration of urolithin A rescued mitophagy and alleviated DMD symptoms [124]. In models utilizing mitochondrial ETC inhibitors, AMPK was key in maintaining quality control. AMPK triggered mitochondrial fission via phosphorylation of MFF as a way to initiate mitophagy targeting damaged mitochondrial fragments. AMPK was thereby postulated to couple fission to mitophagy and initiate biogenesis of new mitochondria to replace damaged ones [125]. Despite fragmentation having been greatly observed in pathology (as described in Section 2.3) and subpopulations of mitochondria uniquely built to be shorter in length (as described in Section 3.2), basal levels of fission and fragmentation constantly occur to maintain quality control. Though compounds activating mitophagy have been shown to alleviate disease symptoms, it is worth noting that these compounds often come with dramatic changes to mitochondrial morphology and are often a reflection of high stress. Mitochondria exhibit strong depolarization upon activation of mitophagy and thus it is not necessarily an ideal treatment as it may prove toxic. The constant quality control of removing damaged mitochondria via initiating fission towards mitophagy gives the populational average appearance of homogenous mitochondria [123].

## 4. Discussion

### 4.1. The Advantages of Mitochondrial Heterogeneity

Mitochondrial heterogeneity can be of great advantage to a cell, most significantly because it allows for metabolic flexibility. In the liver, the ability to properly store excess FFA in the form of lipid droplets protects from hepatic insulin resistance [127]. The ability to properly synthesize lipid droplets has been observed as a unique trait to PDM induced by PLIN5 expression [80,128,129]. While the mitochondria’s ability to increase heterogeneity as a dynamic system is beneficial to cellular plasticity and resilience, the loss or breakdown of mitochondrial heterogeneity can be just as significant. Under a high-fat diet, loss of PLIN5 in hepatocytes protected from steatosis but resulted in hepatic damage and inflammation. Mitochondrial heterogeneity acts as a protective mechanism against lipotoxicity [128,130,131]. Breakdown in heterogeneity ultimately results in an inability to reverse disease states, preventing protective mechanisms from acting. Dynamic heterogeneity in response to a state of cellular injury reveals one of the most beneficial contributions of mitochondrial heterogeneity, allowing for cellular flexibility in respect to both fuel utilization and even in stress-response pathways. While there are numerous instances where mitochondrial heterogeneity is beneficial, there are also instances where heterogeneity can become disadvantageous (as discussed in Section 2).

The ultimate question in mitochondrial heterogeneity is when and how can this typically beneficial process become harmful to a system? Heterogeneity represents a two-sided process, where smaller amounts of heterogeneity can provide a cell with protection and plasticity, yet higher amounts of heterogeneity can lead to irreversible disease onset. Pathology results from the accumulation of dysfunctional or maladaptive mitochondria through promoting processes that enrich heterogeneity, as discussed in Section 2.

### 4.2. Challenges and Considerations to Be Made When Studying Intracellular Heterogeneity

While there is significant evidence that different mitochondrial subpopulations may be present in certain tissue types, dissecting this heterogeneity can prove to be challenging. Mitochondrial heterogeneity has been attributed to different biochemical properties including respiratory or enzymatic activity, membrane potential, morphology, calcium levels, etc., and while these traits may fluctuate in the same direction across different cell types, their functional heterogeneity may not be the same. Often the greatest question is whether or not the observed phenotypes are due to background noise as a result of technical difficulties in separating individual mitochondrial populations [2,8,96,107]. Characterization of mitochondrial heterogeneity thus becomes exceptionally difficult when a portion of measured heterogeneity may stem from the noise generated during isolation, data acquisition, and data analysis. For example, in the case of measuring mitochondrial heterogeneity using single-plane confocal microscopy, a large variance can be generated since not all mitochondria are localized in the same focal plane. When mitochondria are not in the same focal plane, the results exclude mitochondria that are elongating towards a different axis or are covered by another organelle. This potential noise can only be properly assessed with 3D reconstruction at 22 °C, where mitochondrial movement is slowed down when compared to traditional measurement temperatures of 37 °C [4]. Imaging challenges related to the assessment of mitochondrial heterogeneity can arise using electron microscopy (EM) as well. Due to mitochondria being randomly oriented, tissue sectioning for EM results in the cross-sectioning of mitochondria at different angles within a cell. These mitochondria will appear small and round when imaged, generating imaging noise and making them difficult to distinguish from actually fragmented mitochondria in 2D EM. Renal tubule cells make an excellent model for studying mitochondrial fragmentation, as a significant portion of their mitochondria lines up perpendicular to the basement membrane. The unique mitochondrial alignment thereby decreases imaging noise and allows for more viable quantification of fragmented mitochondrial subpopulations [8].

Imaging noise is not exclusive to mitochondrial structure, and similar challenges may be observed in the study of membrane potential. In islet studies, in particular, mitochondria in different focal planes may show reduced membrane potential, thereby excluding some functional roles where membrane potential is relevant for glucose stimulation in islet populations [5]. Alternatively, noise may be generated in the processing leading up to imaging, specifically during the mitochondrial isolation process, as in the separation of brown adipocyte PDM and cytoplasmic mitochondria. Although the isolation method may have impacted the characterization of PDM and cytoplasmic mitochondria, Benador et al. verified the unique population traits by overexpression of the mitochondria–lipid droplet tethering protein to highlight biochemical differences observed in the mitochondrial subpopulations [80,81].

The analysis of mitochondrial diversity has important implications for the diagnosis and treatment of various mitochondrial disorders. The processing and imaging techniques being used for these studies have a major role in the impact and discovery of these studies. Advancement in mitochondrial isolation, analysis, and imaging techniques will thus open promising avenues for utilizing mitochondrial heterogeneity in clinical research.

### 4.3. Remaining Questions in the Field

While there is a significant amount of research outlining the different varieties of mitochondrial heterogeneity, there is little consensus on the origins of heterogeneity and the reasons why cells and tissues adopt heterogeneous subpopulations. Is heterogeneity a product of random changes in mtDNA? Is it a product of protein or RNA transport? Are other organelles involved in the origins of mitochondrial heterogeneity? And is there an energy cost in creating subpopulations? These questions remain largely unexplored, though initial studies show that the answers to these questions may be specific to tissue type, adding to the difficulties of understanding mitochondrial heterogeneity. Another fundamental question that is not yet fully understood is whether one group of mitochondria can become another group of mitochondria. Is heterogeneity created from new subsets of mitochondria or do existing subsets adapt and change into new subpopulations? The field of mitochondrial heterogeneity is growing in importance and dissecting this unique portion of mitochondrial biology will further the diagnosis, understanding, and treatment of many metabolic disorders.

## 5. Conclusions

In the past, mitochondrial subpopulations were harder to distinguish, but the improvement in technology made it feasible. While the sources, causes, and functions of mitochondrial heterogeneity are numerous, mitochondrial heterogeneity remains in a state of duality within biology. Heterogeneity can be highly advantageous for cellular function but there exist many disease states where heterogeneity can exacerbate pathology, transitioning from a protective mechanism to a factor that poses challenges to treatment.

## Figures and Tables

**Figure 1 biology-10-00927-f001:**
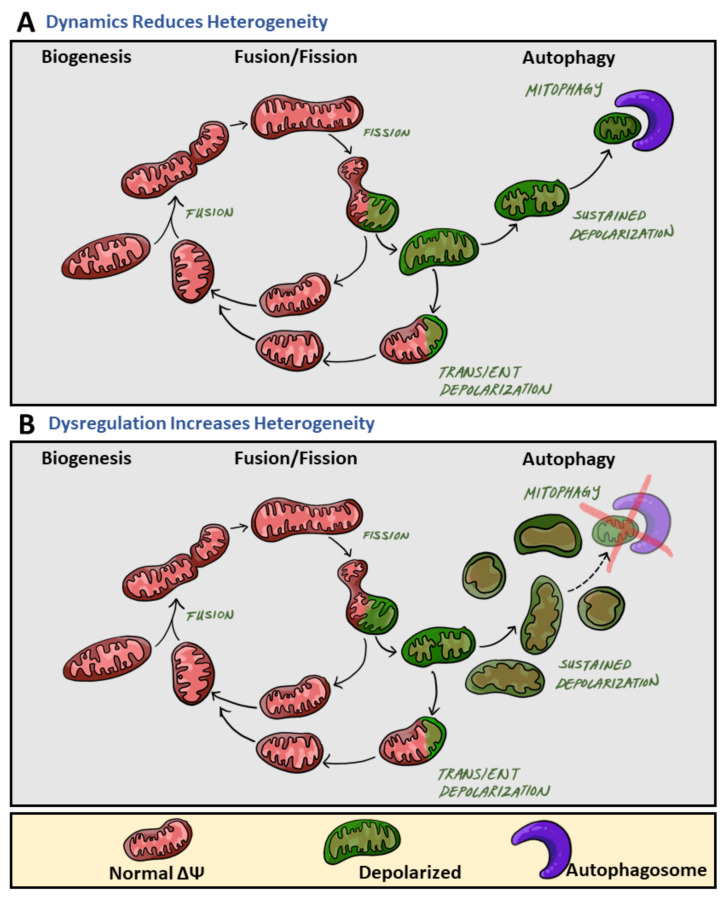
The role of mitochondrial dynamics in mitochondrial heterogeneity. During the mitochondrial life cycle, mitochondrial heterogeneity primarily occurs at three main phases (from left to right): biogenesis, fusion/fission, and autophagy of mitochondria (mitophagy). (**A**) While increases in mitochondrial heterogeneity are beneficial to cellular health, sustained activity of the mitochondrial life cycle reduces mitochondrial heterogeneity. Specifically, fusion events result in the contents of two different mitochondria mixing, ultimately resulting in their contents becoming the same, equilibrating the mitochondrial population within the cell, and thereby decreasing intracellular heterogeneity. After fission, one daughter mitochondrion will depolarize (green) while the other will maintain normal membrane potential (red). Depolarized mitochondria will undergo one of two fates: either a transient depolarization will occur and they will slowly regain their membrane potential to fuse again (green mitochondria back to red), or they will remain depolarized and undergo mitophagy (the mitochondria will remain green). Mitophagy events remove depolarized mitochondria from the mitochondrial population, thereby reducing intracellular membrane potential heterogeneity. (**B**) Dysregulation and blockades within the mitochondrial life cycle lead to increases in heterogeneity. Impaired clearance of mitochondria with sustained depolarization, via mitophagy, increases the pre-autophagic pool. Increasing the pre-autophagic pool size increases intracellular heterogeneity, and the mitochondrial population now transitions from a population with a common membrane potential (red) to one with multiple membrane potentials (red and green).

**Figure 2 biology-10-00927-f002:**
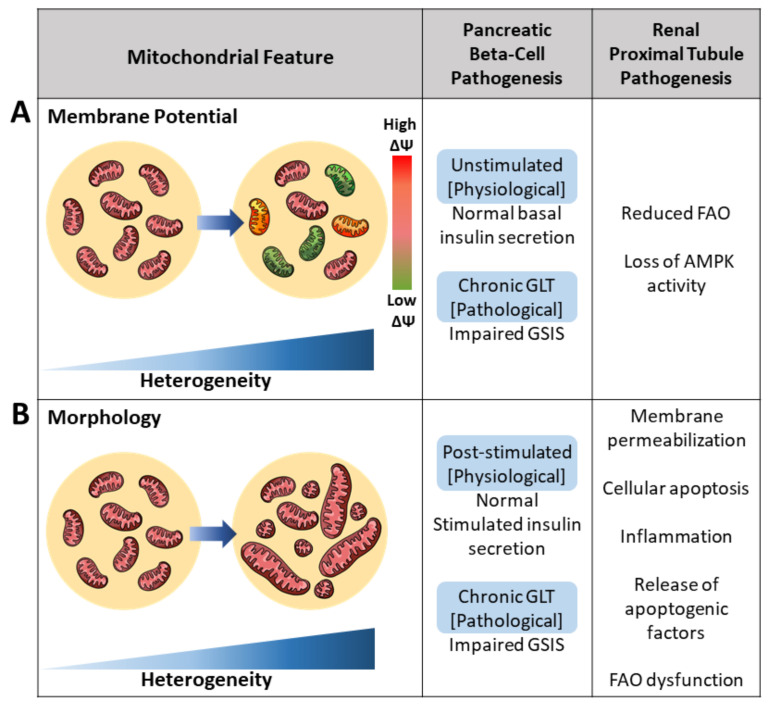
Pathogenesis of the pancreatic beta-cell and renal proximal tubule cell induced by increased mitochondrial heterogeneity. Mitochondrial heterogeneity under the pathological state, such as glucolipotoxicity (GLT), yields various functional readouts. (**A**) Increased mitochondrial membrane potential (ΔΨ) heterogeneity has been associated with various metabolic defects in both the pancreatic beta-cell and renal proximal tubule cell. Intracellular heterogeneity is observed by an increase in differences in ΔΨ. Reduced ΔΨ is presented as green and increased ΔΨ is presented as red. Mitochondria of normal ΔΨ are presented as pink. (**B**) Architectural heterogeneity yields mitochondria varying from short to long mitochondrial lengths as a result of varying fusion and fission rates. Increased heterogeneity in mitochondrial morphology, specifically increased short mitochondria due to reduced fusion, is associated with reductions in impaired glucose-stimulated insulin secretion (GSIS) in pancreatic beta-cells and reductions in renal fatty acid oxidation (FAO).

**Figure 3 biology-10-00927-f003:**
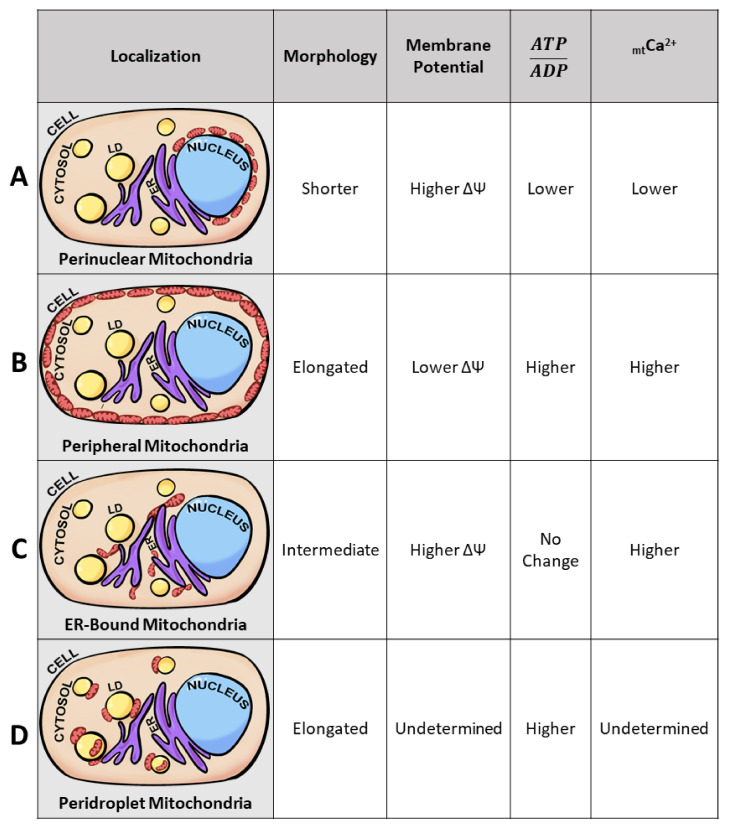
Mitochondrial subpopulations reveal unique mitochondrial traits. Diagram of a typical cell containing nuclei (blue), lipid droplets (LD) (yellow), endoplasmic reticulum (ER) (purple), cytoplasm (peach), and mitochondria (red ovals). Proposed model of distinct mitochondrial subpopulations observed across several cell types. Mitochondria of specific subcellular localizations have unique traits demonstrating that not all mitochondria are homogeneous under physiological states. Based on previous studies [4,23,75,77,80,82,83,84,85,86,87] we observed the presence of 4 mitochondrial subpopulations: perinuclear, peripheral, ER-bound, and peridroplet mitochondria. The 4 mitochondrial subpopulations have been observed to exhibit differences in morphology, membrane potential, ATP/ADP ratios, and mitochondrial calcium (_mt_Ca^2+^) levels. Mitochondrial length is dictated from longest length (elongated), intermediate length, and shortest length (shorter). Higher substrate concentrations are dictated as higher, and reduced substrate concentrations are dictated as lower in the table according to mitochondrial subpopulation. (**A**) Perinuclear mitochondria, localized to the nucleus, display higher membrane potential and reduced mitochondrial length. (**B**) Peripheral mitochondria, bordering the plasma membrane, display increases in mitochondrial length. (**C**) ER-bound mitochondria, tethered to the endoplasmic reticulum, yield higher mitochondrial calcium levels. (**D**) Peridroplet mitochondria, attached to lipid droplets, display elongated mitochondria compared to those in the cytosol.

**Figure 4 biology-10-00927-f004:**
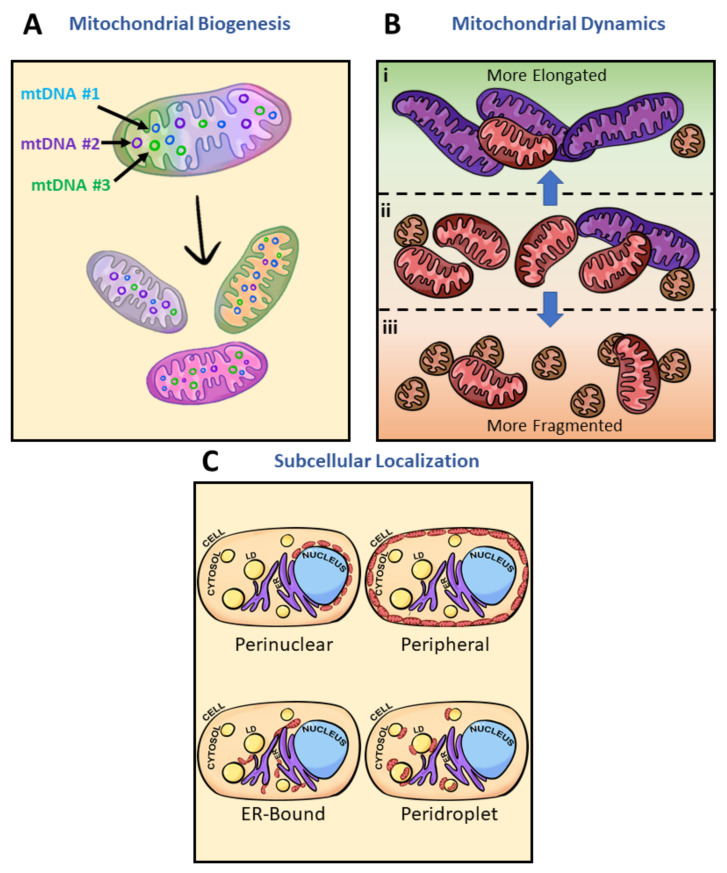
Inducers of mitochondrial heterogeneity. Potential sources of mitochondrial heterogeneity include mitochondrial biogenesis, architectural adaptation, and subcellular localization. (**A**) Mitochondrial biogenesis produces new mitochondria from preexisting ones (tricolor mitochondria); mitochondria which may have mtDNA sequence variation, and can induce variation in mitochondrial protein content or expression. Variations in mtDNA and copy number are represented by the circles of various colors within the mitochondria. Every single color (purple, blue, and green) depicted represents one version of mtDNA or expression pattern. As new mitochondria are made, the content within the new mitochondria will often not be perfect copies of the original mitochondria. Each different daughter mitochondria are of a different singular color to emphasize the dominance of one mtDNA. (**B**) Mitochondrial architectural adaptation often occurs in response to changes in environmental factors such as nutrient levels, oxidative stress, growth hormones, cell signaling, or pathogenicity. Mitochondria at basal state (ii) consist primarily of intermediate-sized mitochondria (red). The highly dynamic organelle has been observed to (i) elongate (purple) much of its population under starvation and (iii) fragment (orange) under nutrient excess. Adaptive mechanisms leading to changes in mitochondrial architecture thereby induce heterogeneity. (**C**) Mitochondrial localization within a single cell and unique traits associated with those regions represent another way heterogeneity is present in physiological states. Mitochondrial heterogeneity and metabolic flexibility are a product of the dramatic differences in mitochondrial phenotypes due to their cellular positions. Perinuclear mitochondria localize near the nucleus, peripheral mitochondria border the plasma membrane, ER-bound mitochondria are tethered to the ER, and peridroplet mitochondria are attached to lipid droplets. Diagram of a typical cell containing nuclei (blue), lipid droplets (LD) (yellow), endoplasmic reticulum (ER) (purple), cytoplasm (peach), and mitochondria (red ovals). For further discussion, see Figure 3.

**Figure 5 biology-10-00927-f005:**
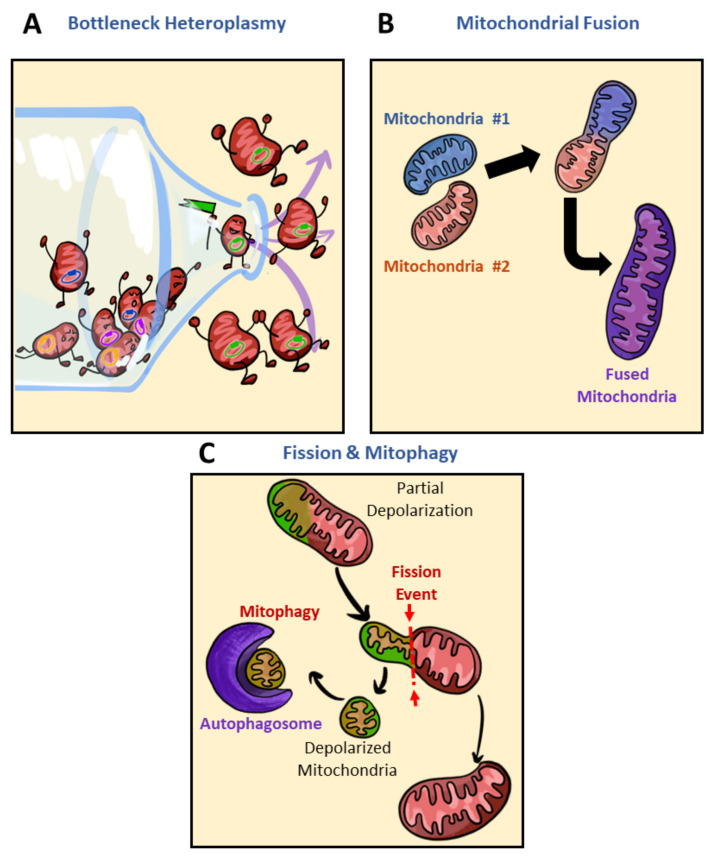
Reducers of mitochondrial heterogeneity. Potential sources of reducing mitochondrial heterogeneity include bottlenecking mitochondrial heteroplasmy, mitochondrial fusion, and mitophagy. (**A**) Though mitochondrial biogenesis has the potential to increase mitochondrial heterogeneity, in cases where the selection of mtDNA experiences a bottleneck effect mitochondrial heterogeneity is reduced. In this cartoon, the bottle containing various mtDNA (yellow, purple, blue, or green) is filtered to only one expression pattern of mtDNA, forcing homogeneity. Through mtDNA bottlenecking, daughter mitochondria of a biogenesis event can lose their heteroplasmic content in favor of one dominating mtDNA sequence variance (mitochondria with the green circle), thereby creating mitochondria that lack genetic diversity and reducing heterogeneity. (**B**) Mitochondrial fusion, though a normal part of the mitochondrial life cycle, can result in a decrease in mitochondrial heterogeneity. When two mitochondria with differing mitochondrial components fuse, their contents are combined, thereby reducing the number of genetic differences between mitochondria in the population. (**C**) Mitochondrial fission and mitophagy is another method by which mitochondrial heterogeneity can be decreased. Through mitochondrial fission, one mother mitochondrion is separated into two, with differences in their membrane potential, one depolarized mitochondrion (green) and the other hyperpolarized or of normal membrane potential (red). Removal of depolarized mitochondria reduces population heterogeneity by reducing the pool of mitochondria with sustained depolarized membrane potential. Polarized mitochondria (red); depolarized mitochondria (green); autophagosome (purple).

## Data Availability

Not applicable.

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
