# Peer review of "Mitochondrial Heterogeneity in Metabolic Diseases"

_biology, 2021, doi:10.3390/biology10090927_

Round 1

Reviewer 1 Report

This review albeit succinct, aims to describe a complex subject. The theme is very interesting and novel but it is somewhat difficult to follow, especially for a reader that is not familiar with the specific topic. Thus, this review would benefit from several clarification and some additions. A major issue is the figures, lacking explanation and detailed legends which makes them difficult to understand.

Specifically;

  1. In the introduction 1. , nothing is mentions about the role of mitochondria in energy production and metabolism under normal circumstances. 1.1 has does not include any reference
  2. The title implies “metabolic diseases” but no inborn error of metabolism linked to aberrant mitochondrial morphology (ie DNM1L, LHON,ATAD3 etc etc etc) is mentioned.
  3. There is no mentioning about the effect compounds/ drugs with the exception of metformin attempting to normalize morphology and mitophagy (bezafibrate, betaine, rapamycin etc.)
  4. Fig 1. Could be expanded to include defective fission fusion i.e (c ) (d). There is no explanation about the color. Does green stand for depolarization?
  5. Figure 2 – what kind of cell is this? Fat cell? Is yellow a fat droplet? Are mito red ? What the does (+) mean? Elevated or present ? Does ( –) mean decreased or absent lacking? It would be nice also to see real micrographs (with permission ).
  6. Fig 3 is very hard to understand without proper figure legend and a color codes. Are the small circles mtDNA? –What does the different color mean –variants? again is yellow a fat droplet ? What is the difference between middle up and middle down? There is no designation (abcd..)
  7. What is the meaning of fig4? There is no explanation do designation (abc)

Minor comment row 191- underline?  

Reviewer 2 Report

The description of mitochondrial morphological changes under physiological conditions, with the latest findings and four categories, is very forward-thinking and interesting. This concept is an area of mitochondrial research that has been assumed to be true but has not been elucidated, and this review has the potential to advance much research. At the end of the book, there is also a research methodology on mitochondrial heterogeneity that will intrigue the reader. The authors have focused on only two organs, the pancreas and the liver, but we judge this to be a problem with few revisions.

Major comment

It is necessary to clarify whether each sentence is about heterogeneity "within cells" or "between cells". These are different conceptions. Overall, the article is about intracellular heterogeneity, but there seems to be a mixture of papers cited that focus on intracellular mitochondrial dynamic heterogeneity and those that do not. As a conclusion to each sentence, it would be easier to understand if you separate whether you are talking about intracellular or intercellular mitochondria.

Minor comments

"Heterogeneity" is easy to identify as an illustration, so you can add a simple explanatory illustration in the sentencing section for pancreas and kidney.

P 6, 2-2

Overall, this review by the authors is a summary of brand-new scientific research findings rather than a review of past concepts, and in some parts, there is a lack of scientific knowledge due to a lack of research. Therefore, I think it is worth reading this review to introduce the part where the molecular mechanism is understood. You move on to the topic of mitochondrial fission and fusion in the middle, but the group of molecules that carry out these processes has been identified to some extent. There are many reviews, and fusion-fission has been studied extensively in the mitochondrial research community. Inserting this section would make this review even better. (Archer SL. Mitochondrial dynamics--mitochondrial fission and fusion in human diseases. N Engl J Med. 2013;369(23):2236-2251. doi:10.1056/NEJMra1215233)

P 6, 2-2

Given that the authors are focusing on intracellular heterogeneity in the context of the second half of the paper. It's a very difficult question, but I don't think many studies of in vivo fission-fusion discuss heterogeneity of mitochondrial dynamics. These show insufficiency or recovery as a result of mitochondria throughout the cell being " unbalanced" toward fission or fusion. In other words, these papers are evaluating whole-cell mitochondria, not intracellular mitochondria. "One-cell" mitochondria over-fission or over-fusion is a frequently pathological condition. The impact of mitochondrial fission and fusion in only specific regions of the cell on pathology has not been investigated because there are no models.

However, since changes in mitochondrial morphology certainly affect mitochondrial heterogeneity, why not introduce the significance of fission-fusion as one of the factors that affect heterogeneity? Alternatively, you may want to rewrite the sentence as "the result of an increased rate of fission" instead of "the result of fission".

P 12, 3-5

Research on individual mitochondria and heterogeneity within mitochondria is most popular in the area of mtDNA. Several models have been proposed, and these should be presented in the paper. (Stewart JB, Chinnery PF. The dynamics of mitochondrial DNA heteroplasmy: implications for human health and disease. Nat Rev Genet. 2015;16(9):530-542. doi:10.1038/nrg3966)

Round 2

Reviewer 1 Report

The revised manuscript is much improved. The f8gures are clear with legends. I have no comments.